# The Role of the Immune System in Pathobiology and Therapy of Myocarditis: A Review

**DOI:** 10.3390/biomedicines12061156

**Published:** 2024-05-23

**Authors:** Cristina Vicenzetto, Andrea Silvio Giordani, Caterina Menghi, Anna Baritussio, Maria Grazia Peloso Cattini, Elena Pontara, Elisa Bison, Stefania Rizzo, Monica De Gaspari, Cristina Basso, Gaetano Thiene, Sabino Iliceto, Renzo Marcolongo, Alida Linda Patrizia Caforio

**Affiliations:** 1Cardiology and Cardioimmunology Laboratory, Department of Cardiac, Thoracic, Vascular Sciences and Public Health, University of Padova, 35128 Padova, Italy; cristina.vicenzetto@aopd.veneto.it (C.V.); renzo.marcolongo@gmail.com (R.M.); 2Cardiovascular Pathology, Department of Cardiac, Thoracic, Vascular Sciences and Public Health, University of Padova, 35128 Padova, Italygaetano.thiene@unipd.it (G.T.)

**Keywords:** myocarditis, immune system, immunosuppressive therapy, autoimmune disease, systemic immune-mediated disease, drug repurposing

## Abstract

The role of the immune system in myocarditis onset and progression involves a range of complex cellular and molecular pathways. Both innate and adaptive immunity contribute to myocarditis pathogenesis, regardless of its infectious or non-infectious nature and across different histological and clinical subtypes. The heterogeneity of myocarditis etiologies and molecular effectors is one of the determinants of its clinical variability, manifesting as a spectrum of disease phenotype and progression. This spectrum ranges from a fulminant presentation with spontaneous recovery to a slowly progressing, refractory heart failure with ventricular dysfunction, to arrhythmic storm and sudden cardiac death. In this review, we first examine the updated definition and classification of myocarditis at clinical, biomolecular and histopathological levels. We then discuss recent insights on the role of specific immune cell populations in myocarditis pathogenesis, with particular emphasis on established or potential therapeutic applications. Besides the well-known immunosuppressive agents, whose efficacy has been already demonstrated in human clinical trials, we discuss the immunomodulatory effects of other drugs commonly used in clinical practice for myocarditis management. The immunological complexity of myocarditis, while presenting a challenge to simplistic understanding, also represents an opportunity for the development of different therapeutic approaches with promising results.

## 1. Introduction: Myocarditis, Hard to Suspect and Tricky to Diagnose

Myocarditis is an inflammatory disease of the myocardium; this nosological entity encompasses a heterogeneous group of diseases characterized by variable clinical presentations and etiologies. 

The most updated and widely endorsed definition of myocarditis is reported in the 2013 consensus document of the European Society of Cardiology (ESC) Working Group on Myocardial and Pericardial Diseases [1]. In this statement, myocarditis and inflammatory cardiomyopathy (if associated with myocardial dysfunction) are not defined just as a generic inflammatory state of the myocardium; instead, specific histological, immunohistochemical and immunological criteria are outlined to achieve a standardized definition of the disease, aiming to address a longstanding challenge in disease classification and diagnosis, which has always been present throughout the history of this condition.

The term “myocarditis” was introduced in 1837 by Jean Cruveilhier, a French pathologist who described inflammation and necrosis within the hearts of individuals who died of rheumatic fever [2]. However, it was not until 1982 that a modern conceptualization of the disease began to emerge, thanks to the work of Woodroof JF et al. who proposed an etiopathogenetic classification system for myocarditis [3]. The advancement in endomyocardial biopsy (EMB) techniques and cardiovascular pathology played a pivotal role in this progress. Not only did these advancements enable the identification of inflammatory cell infiltration in the myocardium, as established by the classic Dallas criteria [4], but they also provided the means to further define this condition.

Recognition of a non-ischemic inflammation in the myocardium has prompted relevant inquiries into characterizing the disease further. Key questions have emerged, including the following: How are the inflammatory infiltrates defined and classified? What factors contribute to myocardial inflammatory states? And to what extent do these factors impact patients’ prognosis and treatment strategies? 

In the 2013 ESC consensus statement, quantitative criteria for pathological characterization were proposed: in EMB examinations, ≥14 leucocytes/mm^2^ should be present, including up to 4 monocytes/mm^2^, along with the presence of ≥7 CD3 positive T-lymphocytes/mm^2^. This recommendation was a steep advance after the publication of the World Health Organization classification of cardiomyopathies in 1995 [5], which lacked established quantitative criteria for EMB analysis, potentially resulting in significant heterogeneity in disease diagnosis.

In the 1995 WHO Classification of Cardiomyopathies [5], inflammatory cardiomyopathy is defined as a specific cardiomyopathy and described as an “inflammatory disease of the myocardium associated with cardiac dysfunction”; the role of virus-negative and/or virus-positive inflammatory cardiomyopathy in the onset and progression of dilated cardiomyopathy (DCM) has recently gained growing attention, due to the emerging etiological treatment options for both infectious and autoimmune/immune-mediated forms [6].

Immunohistochemistry (IHC) emerged as a pivotal technique for EMB analysis in the 2011 consensus statement by the Association for European Cardiovascular Pathology and the Society of Cardiovascular Pathology [7]. IHC was deemed useful for identifying and phenotyping inflammatory infiltrates in EMB, with a list of principal antibodies for immunophenotype characterization provided (namely CD45, CD45RO, CD3, CD20, CD4, CD8 and CD68/PGM1). However, attempts to establish clear cutoff points for inflammatory cellular infiltration and precise criteria for IHC use were based on relatively limited data [8]. Given the contentious nature of several pathological aspects of myocarditis, as highlighted by recent research, further studies are warranted to address these controversies and refine diagnostic criteria [9,10].

To further complicate matters, EMB may not always be feasible for diagnosing myocarditis. In clinical practice, EMB should be considered for each patient in whom myocarditis is suspected, following the exclusion of other potential cardiac or extracardiac diseases that could account for the symptoms and imaging findings, notably coronary artery disease [1]. Nonetheless, EMB can be omitted, at least initially in the clinical evaluation, in the absence of any markers of worse prognosis and when specific etiological therapy is deemed unnecessary [11]. In such a scenario, a definition of clinically suspected myocarditis can be established when a patient presents with a consistent clinical profile. This profile may exhibit considerable heterogeneity, encompassing symptoms ranging from infarct-like acute chest pain to acute or chronic heart failure (HF), unexplained ventricular arrhythmias, to cardiogenic shock; additionally, the diagnosis of clinically suspected myocarditis requires the fulfilment of at least one diagnostic criterion (Figure 1). 

Cardiac magnetic resonance (CMR) has emerged as a pivotal non-invasive diagnostic technique to support the clinical suspicion of myocarditis, due to its ability to provide detailed insights into myocardial tissue characterization. The recently updated Lake Louise criteria [12] require the coexistence of myocardial edema (one T2-based criterion) and non-ischemic myocardial injury (one T1-based criterion) to suspect myocardial inflammation. However, it should be noted that such criteria have only been validated in a relatively small cohort of patients with histologically proven myocarditis [13]. Therefore, further studies are required to better define the diagnostic yield of CMR in inflammatory cardiomyopathy. Additionally, according to the literature, the sensitivity of CMR is significantly influenced by the clinical presentation of myocarditis, potentially attributed to variations in the extent and nature of myocardial injury (e.g., cellular necrosis or apoptosis). Specifically, CMR demonstrates its highest sensitivity when myocarditis presents with infarct-like chest pain (80%), whereas its sensitivity is lower in arrhythmic (57%) and HF (40%) presentations [14].

## 2. Etiology: Infectious Versus Non-Infectious Myocarditis

The clinical spectrum of myocarditis exhibits significant heterogeneity, which can be partially attributed to the wide range of etiological factors contributing to the disease (see Figure 2).

Acute myocarditis can be induced by a multitude of infectious agents, primarily viruses, though bacteria and parasites may also play a role. Geographic variability in the etiology of myocarditis is observed in relation to demographic factors, with viral etiologies prevailing in developed countries, while Central and South America exhibit a higher incidence of Chagas disease, caused by the protozoan Trypanosoma Cruzi [15]. Historically, enteroviruses and adenoviruses were recognized as the predominant etiological agents of viral myocarditis. However, the implementation of viral Polymerase Chain Reaction (PCR) in EMB samples has led to an increased detection of Parvovirus B19 (PVB19) and Human Herpesvirus 6 (HHV-6) [17]. Additionally, other herpesviruses, including Epstein-Barr virus (EBV) and Cytomegalovirus (CMV), have been implicated in the development of myocarditis. Notably, up to 30% of patients with viral myocarditis have been reported to harbor mixed infections [18]. Viruses can be classified based on their tissue tropism as follows (see Figure 2): (a) primary cardiotropic, directly infecting the cardiac myocytes and susceptible to host immune clearance (e.g., adenoviruses and enteroviruses); (b) vasculotropic, targeting endothelial cells (e.g., PVB19) [19]; (c) lymphotropic, capable of latent persistence in the myocardium for years (e.g., HHV6, CMV, EBV); (d) cardiotoxic, inducing myocardial inflammation through immune system activation (e.g., Hepatitis C virus; HIV; A and B Influenza viruses) [16]. It is important to distinguish between myocarditis triggered by viral infection and myocarditis associated with infection. In the former, myocardial damage arises directly from viral replication, whereas in the latter, molecular mimicry between viral antigens and myocardial proteins elicits an immune response targeting myocardial self-antigens [20]. Therefore, the detection of viral agents in the myocardium constitutes only the first diagnostic step in viral myocarditis assessment. To date, no definitive evidence of biopsy-proven myocarditis caused by SARS-CoV-2 infection or vaccination has been found. Conversely, a study demonstrated that SARS-CoV-2 vaccination is safe for patients with prior myocarditis; additionally, this study showed an absence of increased risk of myocarditis relapse after SARS-CoV-2 infection in the same patients’ population [21].

A multiparametric assessment of the viral type and virulence should be performed before considering therapeutic interventions. However, in cases where no virus is detected in EMB, leading to a diagnosis of virus-negative myocarditis, and after excluding all other potential exogenous etiological agents, the disease is presumed to be autoimmune or immune-mediated. Virus-negative myocarditis may present either as an isolated autoimmune disease or within the context of a systemic immune-mediated disease (SID), where extracardiac involvement holds clinical significance. The definition of organ-specific autoimmunity is based on the Rose–Witebsky criteria [22], with inflammatory cardiomyopathy fulfilling various major and minor criteria (see Table 1).

Although cardiac involvement in SIDs is often underestimated or overlooked, it is a common occurrence and serves as an established marker of adverse prognosis that should be recognized to intensify treatment strategies [32,33,34].

A recently defined entity is ICI-related myocarditis (ICIM), an uncommon (0.04–1.14%) yet serious and potentially fatal (with a reported mortality risk of 25–50%) adverse event of Immune checkpoint inhibitors (ICIs). ICIs are monoclonal antibodies targeting immune response receptors, such as CTLA-4 (e.g., ipilimumab), PD-1 (e.g., pembrolizumab), PD-L1 (e.g., avelumab). Due to its peculiar aggressiveness, especially in frail oncological patients, prompt recognition and treatment of ICIM are crucial for successful disease management, which typically involves providing hemodynamic support and administering high-dose intravenous steroid therapy [35].

Few data are currently available on biopsy-proven ICIM cases. In a single-center cohort study including 28 patients with clinically suspected ICIM, EMB showed signs of myocarditis/inflammation in 18 cases. ICIM patients commonly presented with dyspnea and weakness, and less frequently with symptoms of heart failure; additionally, ECG abnormalities, non-sustained ventricular arrhythmias and troponin elevation were reported. Patients’ characterization through EMB had significant therapeutic implications, since a substantial proportion of patients with low-grade myocardial inflammation were able to continue ICI treatment [36].

Regardless of etiology, the main classification of myocarditis is histological: EMB analysis can identify different types, with lymphocytic myocarditis being the most common. However, it can also reveal rare forms such as eosinophilic, sarcoid, polymorphic or giant cell myocarditis (GCM). The type of inflammatory infiltrate is of paramount importance for both prognostic and therapeutic considerations, as will be discussed in the following paragraphs.

## 3. Humoral Immunity in Myocarditis

Among the Rose–Witebsky criteria, the presence of circulating heart-specific autoantibodies in both patients and their relatives is noted. Indeed, in myocarditis, the role of humoral immunity is well established. Various types of autoantibodies have been detected in patients with myocarditis and autoimmune DCM, including anti-β1-adrenergic receptor [37], anti-troponin [38] and anti-M2 acetylcholine muscarinic antibodies [39]. Among these, anti-heart antibodies (AHA) have demonstrated particular relevance in the pathogenesis, diagnosis and prognosis of myocarditis [40,41,42]. AHA presence in the serum of patients with myocarditis correlates with a poor prognosis [43] and their detection in asymptomatic relatives of patients with idiopathic DCM serves as a predictive marker for disease development [24,44].

AHA encompass a group of autoantibodies targeting cardiac antigens, including myosin, troponin and other proteins expressed in the heart tissue. It has been demonstrated that AHA may directly compromise myocardial contractility through complement activation and cell-mediated cytotoxicity [43,45], or indirectly contribute to myocardial damage by inducing an inflammatory response [46,47]. 

Detection of AHA typically involves indirect immunofluorescence, which can further differentiate between various antibody patterns: organ-specific, cross-reactive type 1 or partially organ-specific and cross-reactive type 2 AHA. Organ-specific AHA primarily target the α and β isoforms of the myosin heavy chain [40,45,47,48], with the α isoform uniquely expressed in the atria. Loss of tolerance for cardiac myosin may arise from molecular mimicry mechanisms, cross-reaction between myosin and the β1-adrenergic receptor or cell necrosis triggered by infection or other agents. 

Furthermore, recent studies have highlighted the prognostic role of non-organ specific antibodies, such as anti-nuclear antibodies (ANA), in biopsy-proven myocarditis [42]. Additionally, AHA have been detected in other cardiac diseases where autoimmunity is suspected to be a contributing factor, such as arrhythmogenic cardiomyopathy, suggesting an association with disease severity [49]. Further research is warranted to better characterize the precise mechanisms of AHA induction in myocarditis and their presence and role in other cardiomyopathies.

## 4. Immunopathological Processes in Myocarditis

Both innate and adaptive immunity contribute to the pathogenesis of myocarditis. These mechanisms can be targeted through conventional immunosuppressive or immunomodulatory treatments, as well as by repurposed drugs already used to manage myocarditis symptoms with newly recognized immunomodulatory effects. The pathogenesis of myocarditis encompasses infectious or autoimmune/immune-mediated mechanisms, occurring in three distinct phases. In the acute phase lasting 1–7 days, the innate immune response plays a role, potentially triggered by an infectious agent, e.g., a viral infection. One mechanism by which viruses infect cardiomyocytes is through internalization via receptor complexes interaction, especially the coxsackie/adenoviral receptor (CAR) [50]. Internalization of the virus is facilitated by concurrent binding to decay accelerating factor (DAF; also known as CD55), a ubiquitously expressed host protein that inhibits complement activation [50,51]. 

Subsequently, in the subacute phase lasting 1–4 weeks, the adaptive immune response becomes engaged. In a sizable proportion of cases, the inflammatory process resolves spontaneously; however, incomplete resolution of inflammation may lead to a chronic course in a relevant quote of patients (up to 25%). This chronic phase can extend from months to years, ultimately progressing to DCM, end-stage HF and death [52].

Different types of innate and adaptive immune cells participate in the pathogenesis of myocarditis across different stages, including acute infection, subacute immune response and adverse cardiac remodeling. 

Innate immunity comprises humoral effectors such as cytokines, chemokines and complement, as well as cellular effectors like neutrophils, monocytes/macrophages. Following infection, the innate immune response is initiated, involving pattern recognition receptors (PRRs) expressed on the surface of innate immune cells which identify pathogen-associated molecular patterns (PAMPs) and damage-associated molecular patterns (DAMPs) released from damaged cells, subsequently triggering the release of cytokines, chemokines and alarmins [53,54]. 

This cascade recruits inflammatory cells including mast cells, neutrophils, dendritic cells, monocytes and macrophages to the heart. Notably, monocytes and macrophages are the predominant inflammatory cell subsets identified in both human and experimental myocarditis [55]. Monocyte recruitment and migration play crucial roles in myocarditis development. Cardiomyocytes secrete key chemokines such as monocyte chemoattractant protein 1 (MCP-1) and macrophage inflammatory protein-1 alpha (MIP-1α), which promote monocyte migration and contribute to myocarditis via C-C chemokine receptor (CCR) type 2 and CCR5, respectively [56,57]. In a murine model of CVB3-induced myocarditis, increased MCP-1 expression in cardiomyocytes was observed upon CVB3 infection [58]. Additionally, silencing CCR2 using short interfering (si)RNA in experimental autoimmune myocarditis reduced inflammatory monocyte recruitment, thereby attenuating myocardial inflammation and fibrosis [59].

Once at the site of injury, monocytes amplify inflammation by further cytokine expression, including Interleukin (IL)-6 and Tumor necrosis factor (TNF)-α, as well as differentiation into macrophages with different functions [60,61].

Traditionally, macrophages are divided into 2 types according to their activating factors [62]. The first type is classical (M1) macrophages activated by Toll like receptor (TLR) ligands or Interferon (IFN)-γ, which mainly mediate oxidative stress, inflammasome formation and proinflammatory factor secretion (TNF-α, IL-1β, IL-6 and IL-12, among others) to produce co-stimulating factors and chemokines that promote inflammatory cell infiltration and proliferation. They subsequently cause cardiac injury [63]. In contrast, IL-4 and IL-13, as well as IL-10 and Transforming growth factor (TGF)-β, induce alternatively activated M2 macrophages, mainly playing an anti-inflammatory role [64,65,66]. Throughout the different phases of tissue repair, macrophages shift from M1 to M2 phenotype. M1 macrophages in early inflammation produce IL-6, TNFa and IL-1, activating innate immune cells. During resolution, M1 to M2 transition reduces inflammation, increases anti-inflammatory cytokines and clears apoptotic neutrophils and damaged cells [63,64,65]. In BALB/c male mice with CVB3-induced myocarditis, myocardial macrophage phenotype shifts from M1 to M2 around days 7 to 10 during fibrotic repair [67].

In addition to innate immune cells, the pathogenesis of myocarditis involves crucial contributions from adaptive immune pathways, notably T helper 1 (Th1) and T helper 2 (Th2) cells, along with key cytokines. The balance between Th17 and regulatory T cells (Treg) also significantly influences myocarditis progression [68,69,70]. Th1/Th2 paradigm was introduced to categorize CD4+ T cells based on their cytokine secretion profiles. In the clinical phase of myocarditis, a systemic Th1/Th2 imbalance is well-documented [71]. The exact role of Th1 cells in myocarditis pathogenesis is still debated, as they seem to play a dual role, initiating tissue damage while also providing protection against excessive inflammation. The secretion of IL-12 by these cells has been reported to be proinflammatory, since exogenous administration of IL-12 exacerbates heart inflammation in experimental autoimmune myocarditis (EAM) models, which can be blocked in IL-12R knockout mice [72]. Conversely, IFN-γ, produced by Th1 cells, limits disease progression to the chronic second phase of myocarditis, essentially DCM and end-stage HF [72]. Inhibiting IFN-γ using monoclonal antibody or gene inactivation promotes a Th2-type immune response that increases acute myocarditis, pericarditis and DCM in the EAM model [73,74]. This effect may be attributed to IFN-γ ability to activate macrophages, facilitating the eradication of intracellular viruses, and regulating the expansion of activated T lymphocytes [75,76]. Th2 cells and associated cytokines such as IL-4 and IL-13 play a crucial role in severe myocarditis characterized by eosinophil expansion. Susceptibility to experimental autoimmune myocarditis (EAM) in A/J and BALB/c mice depends on the production of an IL-4/Th2-type immune response, which in mice is associated with eosinophils, elevated IgG1 autoantibodies against cardiac myosin and elevated IgE [73,77]. Conversely, IL-13, known for its regulatory effects on macrophage differentiation, mitigates the extent of myocardial inflammation, underlining also for Th2 cells a “dual” action in myocarditis pathogenesis [73]. 

The classical “Th1-Th2” paradigm has evolved to incorporate other distinct T helper cell subsets, notably Th17, which plays a key role in the initiation and progression of immune-mediated diseases [78]. Th17 cells pathological differentiation and signalling are predominantly driven by IL-6 and IL-23 [78]. Inhibition of the transcription factor STAT3, which serves as the principal mediator of IL-6 signalling and Th17 differentiation, was found to impair the development of EAM and to improve heart function during DCM [79]. Additionally, high levels of Th17 cells have been confirmed in human myocarditis and DCM both in peripheral blood and myocardial tissue, and they were associated with a lack of left ventricular (LV) recovery and progression to more advanced HF stages [80]. While the Th17 signature cytokine IL-17A is not essential for myocarditis onset, it is crucial for the progression to HF [81,82].

Despite advancements in both experimental and clinical research, the immunological background of myocarditis remains only partially understood. Further investigation is warranted to discover novel therapeutic approaches targeting specific immune pathways in the clinical management of myocarditis.

## 5. Therapeutic Approaches: Targeting the Immune System to Heal Myocarditis

The 2013 ESC consensus statement recommends immunosuppressive therapy (IT) for selected cases of histologically confirmed virus-negative myocarditis [1], particularly giant cell myocarditis (GCM), necrotizing eosinophilic myocarditis or cardiac sarcoidosis (Table 2). This recommendation has been recently reaffirmed in the 2021 ESC guidelines for the diagnosis and treatment of HF [83].

Evidence regarding the efficacy of IT for treating heart failure in biopsy-proven virus-negative lymphocytic myocarditis is based on randomized clinical trials (RCTs) and metanalysis [28,29,30,85]. Conversely, data concerning the efficacy of IT in other forms of histological myocarditis such as GCM [88], eosinophilic myocarditis [89] and cardiac sarcoidosis [90] primarily come from retrospective observational registries. 

According to the limited number of RCTs focusing on biopsy-proven virus-negative inflammatory cardiomyopathy, IT can improve left ventricular function, alleviate symptoms and increase overall survival among this specific subset of patients with myocarditis [29,91]. With respect to lymphocytic myocarditis, the majority of available data primarily focus on the use of a combination of corticosteroids and a steroid-sparing agent, mainly Azathioprine (AZA) [29,85] or Mycophenolate Mofetil (MMF) [86]. However, the treatment of other histological types, such as GCM, entails a combination of at least three drugs, commonly including Cyclosporine A, together with corticosteroids and a steroid-sparing agent (AZA or MMF) [88]. Indeed, GCM is recognized as “the most fatal of autoimmune diseases” and is characterized by an aggressive clinical course, with a high incidence of fulminant hemodynamic presentation and substantial mortality rate [92]. In cardiac sarcoidosis, steroid therapy is indicated in cases of ventricular dysfunction or arrhythmias [93]. 

When managing myocarditis within the context of SIDs, treatment should encompass standard approaches tailored to address the underlying condition. Additionally, given the adverse prognosis of patients with myocarditis in the context of SIDs, more intensive immunosuppression regimens may also be considered [94].

It is crucial to note that the presence of viral agents in the myocardium should be considered as an absolute contraindication to the use of IT. The Myocarditis Treatment Trial, the first clinical study investigating the effects of IT in addition to Optimal Medical Therapy (OMT) in 111 patients with myocarditis of unknown etiology, did not demonstrate an improvement in terms of survival in IT patients compared to those receiving OMT, despite an overall improvement in left ventricular function [95]. The primary limitation of the study was the unknown etiology of myocarditis: in this trial, EMBs were solely analyzed by histological Dallas criteria, without including a search for viral genome in the myocardial tissue samples. Furthermore, the Myocarditis Treatment Trial lacked sufficient statistical power to detect differences in survival.

In a retrospective analysis by Frustaci et al. [91], the virological and immunological characterization of lymphocytic myocarditis treated with IT revealed a 90% response rate in virus-negative cases, whereas viral genomes were detected in the myocardium of 85% of non-responders. Consequently, updated recommendations suggest the use of IT only after ruling out active infection in EMB by PCR, in the absence of other contraindications [1]. 

The Tailored Immunosuppression in virus-negative Inflammatory Cardiomyopathy (TIMIC) trial [85], a randomized double-blind study conducted in 2009, presented significantly different results. It aimed to assess the efficacy of IT (PDN and AZA) in 85 patients with biopsy-proven virus-negative active myocarditis and chronic HF unresponsive to OMT, compared to a placebo control group. This RCT demonstrated a significant improvement in LVEF (left ventricular ejection fraction) in 88% of patients receiving IT, while 83% of patients in the placebo group experienced progressive LV dysfunction. Furthermore, follow-up EMBs showed the resolution of inflammatory infiltrates among the patients exhibiting improvement with IT. Subsequently, a 20-year follow-up study on the same cohort, expanded upon these results, confirming the efficacy of IT both in the short and long term, even among patients with poor baseline conditions (i.e., severe impairment of left ventricular function) and for preventing relapses in autoimmune myocarditis [29]. Similarly, a RCT on 84 patients diagnosed with DCM and increased HLA expression, despite revealing no difference in a composite outcome comprising death, heart transplant and hospital readmission, demonstrated a significant improvement in LVEF and a reduction in LV volumes among patients treated with IT [96].

The safety and efficacy of tailored IT in all histological types of virus-negative myocarditis has recently been confirmed in a single centre prospective cohort using a propensity weighted approach [31].

Among novel approaches to immune-mediated myocarditis, monoclonal antibodies also represent a promising tool. For instance, Mepolizumab, an anti-IL-5 antibody, has been successfully used to treat patients with forms of eosinophilic myocarditis. Additionally, many other antibodies have been tested in vitro to selectively target pro-inflammatory cytokines or immune cell surface receptors, thereby potentially attenuating myocardial inflammation and preserving cardiac function [97]. Rituximab, an anti-CD20 monoclonal antibody with potent immunosuppressive properties, has also been used in selected cases of virus-negative myocarditis, with promising results [98]. Moreover, also for recent cases of ICIM a biotechnological approach involving the blockade of co-stimulatory proteins CD80/CD86 with Abatacept appears to be efficacious [99,100].

The predictors of IT response in myocarditis are still under research. Exploration of the specific cytokines and molecular pathways, both within myocardium and at the peripheral level, as well as the assessment of genetic predisposition, warrants further investigation.

## 6. Repurposing of Traditional Drugs with Potential Immunomodulant Action for Myocarditis Management 

Given the pathogenic role of the immune system in myocarditis, several attempts have been made to target it for myocarditis treatment with direct immunosuppression, as described above. Nonetheless, several studies demonstrated that drugs with other clinical indications, not initially designed to act on immunity, might have novel molecular mechanisms of action on cells of the immune system, thus supporting the rationale of their application in inflammatory cardiomyopathy. Many drug categories were suggested, such as anti-hypertensive, anti-diabetic, anti-HF and hypolipidemic agents; notably, some of these might be already used in myocarditis patients since they belong to OMT (Figure 3).

Among anti-hypertensive agents, the non-selective β-adrenergic receptor (β-AR) blocker carvedilol, was demonstrated to mediate a cardioprotective role either in models of fulminant autoimmune or viral myocarditis. Its cardioprotective role is exerted by decreasing the troponin and myofilaments degradation through the reduction of the Matrix Metalloproteinase-2 activity, leading to the preservation of cardiac function [101]. Moreover, carvedilol acts also as antioxidant and anti-inflammatory; in particular, the inhibition of β-2AR, expressed on Th1 lymphocytes and antigen presenting cells, leads to a reduced expression of pro-inflammatory cytokines IL-1β and TNF-α and the promotion of the anti-inflammatory cytokines IL-10 and IL-1RA [102,103]. While in murine viral myocarditis carvedilol improved the survival lowering myocardial inflammation and necrosis by lower IL-6 and TNF-α secretion [104]. Another class of anti-hypertensive drugs with proven immunomodulant action in myocarditis models are the Angiotensin-II receptor 1 antagonists (AT1R). In fact, Angiotensin-II (AngII) is upregulated in the heart of EAM models and induces monocytes/macrophages chemotaxis; thus, the use of losartan reduces heart infiltration and ameliorates the disease phenotype [105]. Generally, sartans act as an anti-inflammatory agent in the heart either in innate or acquired immunity, since in models of autoimmune or viral myocarditis they reduce the expression of the Th1 type pro-inflammatory while increasing the Th2 ones, leading to a rebalanced Th1/Th2 ratio [106,107,108]. Notably, sartans might have a specific anti-inflammatory action in myocarditis, since in viral models their cardioprotective role was not linked to any antiviral action [106]. AT1R inhibition contributes also to the prevention of DCM evolution because it lowers the expression of endoplasmic reticulum (ER) stress proteins, thus reducing myocardial apoptosis and fibrosis [107,109,110]. Renin-Angiotensin System (RAS) might be targeted not only through the inhibition of target receptors, but also in lowering the production of the soluble mediators as Ang II by angiotensin converting enzyme (ACE) inhibitions. In particular, captopril or similar, when compared to losartan, has similar mechanisms of action in reducing cardiac hypertrophy, fibrosis and AHA production, despite, for the latest, controversial results being reported [111,112,113]. While by blocking AT1R is possible to decrease chemotaxis of the macrophagic compartment, captopril reduces the effect of the hypersensitivity to anti-myosin antibodies by reducing local inflammatory processes, limiting T cells recruitment to the heart [111]. Captopril may therefore reduce inflammation by inhibiting recruitment of T cells to the heart or by reducing local inflammatory processes. 

Sartans, specifically telmisartan, induce the down-modulation of AT1R, which might drive to the upregulation of peroxisome proliferator activated receptor (PPAR)-γ, whose increased activity is linked to myocarditis improvement [107]. Indeed, the class of PPAR-γ activators, as the antidiabetic drug Pioglitazone, is indicated as beneficial for myocarditis. In particular, PPAR-γ is highly expressed in infiltrating inflammatory cells and its activation is linked to a strong nuclear factor kappa-light-chain-enhancer of activated B cells (NF-κB) inhibition, which leads to a reduced expression of the chemotactic protein MIP-1α and Th1 type cytokines while increasing Th2 type, altogether promoting the shift from Th1 towards Th2 [114,115,116]. In the last decade novel antidiabetic drugs with stunning anti-HF properties have been evaluated in autoimmune myocarditis models, such as Linagliptin and gliflozins. Linagliptin acts by blocking Dipeptidyl peptidase (DPP)-4 and it is approved for diabetes treatment, but, since DDP-4 is expressed also in T lymphocytes, it might be acting also in modulating inflammatory response. In fact, in EAM and ICIM models linagliptin is effective in reducing fibrosis, pro-inflammatory cytokine secretion, as well as the AngII production. In particular, the blockade of DDP-4 inhibits Catepsin to catalyze AngII production [117,118]. Another class of recently approved antidiabetic drugs, which have been rapidly included among the “four pillars” for HF treatment, namely the inhibitors of sodium/glucose cotransporter 2 (SGLT-2) gliflozins, has been studied in EAM models. Despite SGLT-2 is not expressed on cardiomyocytes, gliflozins and particularly canagliflozin, might act on cardiac homeostasis by targeting other proteins involved, as SGLT-1, which is upregulated in EAM models, and its inhibition ameliorates oxidative stress blocking NF-κB [119,120]. Gliflozins have cardioprotective effects also by reducing fibrosis and cardiac inflammation and apoptosis through the following: shifting of macrophages polarization from M1 to M2 by blocking the pro-inflammatory pathway signal transducer and activator of the transcription (STAT)3 activation; inhibition of the NLR family pyrin domain containing 3 (NLRP3) inflammasome leading to a reduced IL-1β and IL-18 production and Th17 cells infiltration [120,121]. Angiotensin receptor-neprilysin inhibitors (ARNI), another type of anti-heart failure drug, may be beneficial in myocarditis inhibiting the differentiation of Th17 cells, as gliflozins, but with a different mechanism of action; in EAM models, they do not block the NLRP3 inflammasome pathway, but they activate natriuretic peptide receptors, expressed on heart infiltrating lymphocytes, which block NF-κB p65 signalling pathway [122]. ARNI have been described as effective in models of viral myocarditis; in particular, in vitro experiments show a reduction in the production of IL-6 and IL-1β after treatment with LCZ696 [123].

Interestingly, the NF-κB pathway is a relevant pathogenetic mechanism of inflammation in myocarditis, its inhibition being involved in several anti-myocarditic mechanisms by blocking Th17 differentiation and infiltration in EAM. Another example of this is the hypolipidemic drug fenofibrate, which is a PPAR-α agonist. PPAR-α is part of the family of PPAR proteins abovementioned and acts on cardiac lipid metabolism [124].

Statins, lipid-lowering drugs known as 3-hydroxy-3-methylglutaryl coenzyme-A (HMG-CoA) reductase inhibitors, have also been reported having immunomodulating roles and several studies in myocarditis models have been described in the last years [125]. In particular, statins can reduce proinflammatory Th1 cytokines production, through NF-κB inhibition, as TNF-α, IFN-γ, IL-6 and IL-2, in favour of Th2 type as IL-10 and IL-4 [126,127,128]. The reduction of TNF-α, statin mediated, leads to ameliorated potassium currents in ventricular cardiomyocytes and to reduced Major histocompatibility complex (MHC) class II and co-stimulatory proteins expression on antigen presenting cells [129,130]. Nonetheless, statins have been recently used, together with intravenous immunoglobulins, to treat three cases of ICIM reaching LVEF improvement and class I NYHA at discharge [131].

## 7. Conclusions and Future Directions

Current guidelines [83] recommend the use of immunosuppressive therapy in selected cases of virus-negative biopsy-proven myocarditis, on top of OMT that should be targeted on the specific patient’s phenotype, e.g., anti-HF drugs in case of left ventricular dysfunction or antiarrhythmic medical therapy in case of hemodynamically stable sustained ventricular arrhythmias [132]. 

Emerging evidence suggests that HF medications, such as β-blockers or ACE-inhibitors, have immunomodulatory effects in addition to their primary ones in murine models [104,110]. These findings, even if obtained mainly in models, raise an intriguing possibility for the management of myocarditis patients: could a novel combination of IT and OMT potentially enhance patients’ response through a synergistic effect of drugs? To answer this question, clinical studies are warranted along with the ongoing research on novel therapeutic approaches. Despite advancements in understanding the mechanisms, the pathogenesis of myocarditis involves an intricate immune network, complicating so far a clear determination of the most relevant factors; therefore, the exact role of non-immunosuppressive/immunomodulating agents in myocarditis in vivo still needs to be ascertained. In conclusion, this review contributes to describe the main etiological, clinical and therapeutic features of myocarditis, underlining the importance of new immunopathological insights for their strong therapeutic potential. In particular, emerging evidence regarding repurposed drugs, as well as novel biotechnological drugs, suggests promising therapeutic options for myocarditis.

## Figures and Tables

**Figure 1 biomedicines-12-01156-f001:**
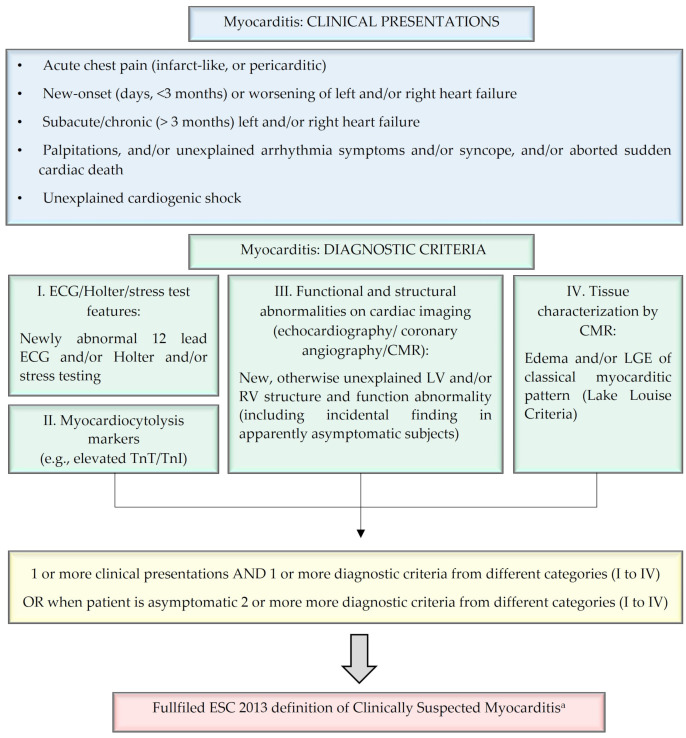
Diagnostic criteria for clinically suspected myocarditis, modified from Caforio et al. [1]. Myocarditis is a highly heterogeneous disease; these criteria were defined in 2013 by the European Society of Cardiology (ESC) Working Group on Myocardial and Pericardial Diseases with the purpose of facilitating clinicians in recognizing myocarditis and addressing selected patients to second- or third-level examinations and possible etiological treatment. ^a^ Clinically suspected myocarditis is diagnosed if ≥1 clinical presentation (upper panel) and ≥1 diagnostic criteria (lower panel) from different categories are present; if no symptoms are reported, ≥2 diagnostic criteria (lower panel) should be met. Legend: CMR: cardiac magnetic resonance; ECG: electrocardiogram; LGE: late gadolinium enhancement; LV: left ventricle; RV: right ventricle; TnI: troponin I; TnT: troponin T.

**Figure 2 biomedicines-12-01156-f002:**
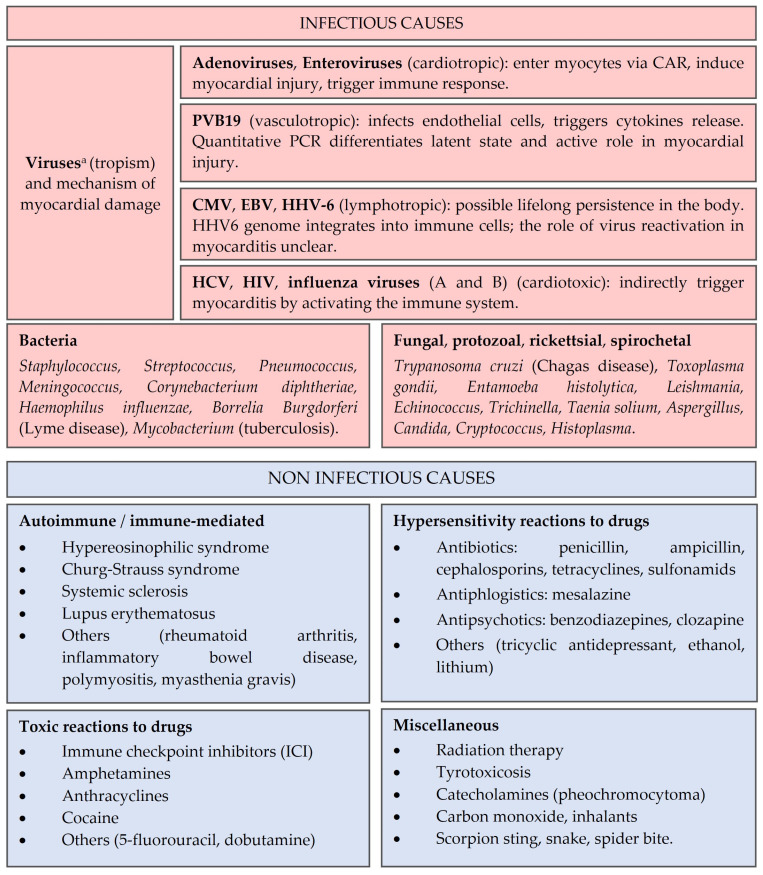
A schematic résumé of possible myocarditis etiologies; adapted from Brociek et al. [15] and from Tschöpe et al. [16]. The list is not exhaustive. ^a^ Viral presence in the myocardium is to be assessed through viral Polymerase Chain Reaction (PCR). Legend: CAR: coxsackievirus and adenovirus receptor; CMV: cytomegalovirus; EBV: Epstein–Barr virus; HCV: hepatitis C virus; HHV-6: human herpesvirus 6; HIV: human immunodeficiency virus; ICI: immune checkpoint inhibitors.

**Figure 3 biomedicines-12-01156-f003:**
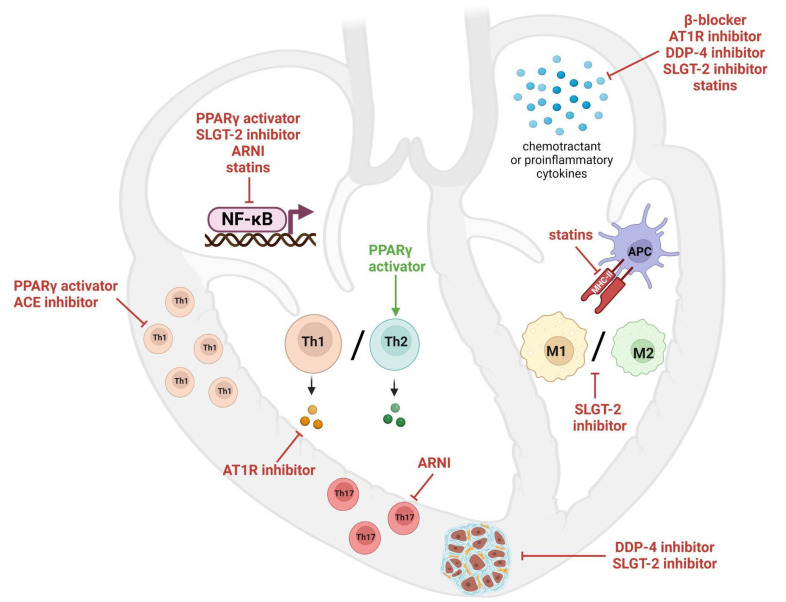
Immunomodulatory effects of repurposed drugs in myocarditis. The figure shows the main pathogenetic mechanisms in myocarditis that are targeted not only by classical immunosuppressive agents, but also by other drugs with a different primary mechanism of action. (PPAR = Peroxisome Proliferator Activated Receptor; SGLT-2 = Sodium/Glucose Cotransporter 2; ARNI = Angiotensin receptor-neprilysin inhibitors; ACE = Angiotensin converting enzyme; AT1R = Angiotensin-II Receptor 1; DDP-4 = Dipeptidyl Peptidase 4; NF-κB = nuclear factor kappa-light-chain-enhancer of activated B cells; Th = T helper lymphocyte; M = macrophage; MHC-II = Major histocompatibility complex class II; APC = antigen presenting cell). Created with BioRender.com.

**Table 1 biomedicines-12-01156-t001:** Rose–Witebsky criteria in inflammatory cardiomyopathy. Adapted from Tschöpe et al. [16].

Criteria	Evidence in the Literature
Presence of immune cell infiltrates and abnormal expression of HLA class II on interstitial cells; Presence of adhesion molecules in the absence of viral genomes in EMB samples from both index patients and family members.	Kindermann et al., 2008 [23]
Presence of circulating heart-specific autoantibodies in patients with inflammatory cardiomyopathy and their relatives.	Caforio et al., 2007 [24] Mestroni et al., 1999 [25] Baritussio et al., 2022 [26]
Animal models of experimentally induced inflammatory cardiomyopathy, with or without a DCM phenotype, after immunization with specific auto-antigen(s).	Li et al., 2006 [27]
Response to immunosuppressive or immunomodulant therapy in patients with virus-negative inflammatory cardiomyopathy.	Escher et al., 2016 [28] Chimenti et al., 2022 [29] Merken et al., 2018 [30] Caforio et al., 2024 [31]

Legend: DCM: dilated cardiomyopathy; EMB: endomyocardial biopsy.

**Table 2 biomedicines-12-01156-t002:** Immunosuppressive therapy protocols in autoimmune/immune-mediated virus-negative myocarditis. Modified from Baritussio et al. [84].

EMB Diagnosis	Treatment	Comments
Lymphocytic myocarditis	PDN, 1 mg/kg/day for 1 month, then gradually tapered and discontinued within 5 months + AZA *, 2 mg/kg d for 6 months MMF, starting with 1 g/day, then increasing to 2 g/day over 4 weeks, and up to 3 g/day if required, in combination with PDN. ** MPDN, i.v. 1 g bolus, for 1 or more days, then, 1 mg/kg/day to be gradually tapered	(Frustaci et al., 2009 [85]; Chimenti et al., 2022 [29]) MMF: Off-label for myocarditis; Second line if intolerance or resistance to AZA (De Luca et al., 2020 [86]); First line in SIDs.
Giant Cell Myocarditis	MPDN, 10 mg/kg i.v. bolus + OKT3, 5 mg/day for 10 days. Then, PDN p.o.: 1 mg/kg/day, then gradually tapered + Cy-A (therapeutic blood range 150–300 ng/mL) and AZA *, starting with mg/Kg/day.	(Cooper et al., 1997 [87])
Eosinophilic myocarditis	PDN p.o., 1 mg/kg die to be gradually tapered in combination with weekly MTX, 7.5–20 mg (alternatively AZA *, 1–2 mg/Kg/day), or MMF, 1–3 g/day Treatment of underlying disease when associated to EGPA. ** MPDN, i.v. 1 g bolus, for 1 or more days, then, 1 mg/kg/day to be gradually tapered	MMF: off-label for myocarditis
Cardiac Sarcoidosis	PDN, 1 mg/kg die to be gradually tapered in combination with weekly MTX, 7.5–20 mg/week (alternatively, AZA *, 1–2 mg/Kg/day), or MMF, 1–3 g/die; If no response, MTX, 15–20 mg/week, in combination with i.v. Infliximab, 5 mg/kg, time 0, at 2 weeks and 4 weeks, then every 8 weeks.	MMF: off-label for myocarditis

Legend: anti-CD3 monoclonal antibody; AZA: Azathioprine; Cy-A: Cyclosporin-A; EGPA: Eosinophilic Granulomatosis with Polyangiitis; EMB: Endomyocardial Biopsy; i.v.: intravenous; MMF: Mycophenolate Mofetil; MTX: Methotrexate; MPDN: Methylprednisolone; p.o.: per os; PDN: Prednisone; SIDs: Systemic Immune-mediated Diseases. * in absence of Thiopurine-Methyl-Transferase mutations. ** fulminant onset.

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
