# Peer review of "The Role of the Immune System in Pathobiology and Therapy of Myocarditis: A Review"

_biomedicines, 2024, doi:10.3390/biomedicines12061156_

Round 1

Reviewer 1 Report

Comments and Suggestions for Authors

Overall reasonable summary.  Good review of data.  Many charts and tables - would have e appreciated a graphic which is easier and more interesring to learn from. 

Author Response

We thank the Reviewer for this comment.  We have transformed Table 1 into Figure 1 to address this concern: we have visually reorganized the information on the spectrum of myocarditis clinical presentations and the criteria for the diagnosis of clinically suspected myocarditis. Similarly, Tables 2 and Table 3, which detailed possible myocarditis etiologies and pathological mechanisms, were reorganized into Figure 2.

Reviewer 2 Report

Comments and Suggestions for Authors

Congratulation.

A nice and comprehensive review.

Author Response

We thank the reviewer for the positive feed-back.

Reviewer 3 Report

Comments and Suggestions for Authors

This is a fine, well-addressed review about myocarditis. In particular, the possible efficacies of heart failure drugs such as beta blocker or ACE inhibitor for myocarditis was concisely documented, which had not been reviewed by other manuscripts. However, there were some points to be addressed.

# I think it would be better to explain a little more about inflammatory cardiomyopathy, a concept that has recently emerged.

# Immunocheckpoint inhibitor myocarditis or covid vaccine induced myocarditis should be documented. 

Comments on the Quality of English Language

No comment

Author Response

Reviewer #3 wrote: This is a fine, well-addressed review about myocarditis. In particular, the possible efficacies of heart failure drugs such as beta blocker or ACE inhibitor for myocarditis was concisely documented, which had not been reviewed by other manuscripts. However, there were some points to be addressed.

  1. I think it would be better to explain a little more about inflammatory cardiomyopathy, a concept that has recently emerged.

Reply: We thank the Reviewer for this comment. We have expanded the concept of the definition of inflammatory cardiomyopathy proposed by the 1995 WHO Classification of Cardiomyopathies, and included an update on therapeutic implications in light of recent advances in myocarditis treatment. Revised text reads as follows:

In the 1995 WHO Classification of Cardiomyopathies [5], inflammatory cardiomyopathy is defined as a specific cardiomyopathy and "inflammatory disease of the myocardium associated with cardiac dysfunction”; the role of virus-negative and/or virus-positive inflammatory cardiomyopathy in the onset and progression of dilated cardiomyopathy (DCM) has recently gained growing attention, due to the emerging etiological treatment options for both infectious and autoimmune/immune-mediated forms (new reference number [6]) (lines 62-67).

  1. Immunocheckpoint inhibitor myocarditis or covid vaccine induced myocarditis should be documented.

Reply: We thank the Reviewer for this comment. We have broadened the discussion on immune checkpoint inhibitors myocarditis (ICIM): we have detailed ICIM possible clinical presentations and histological findings, and highlighted the role of EMB in guiding treatment decision regarding the continuation or discontinuation of ICI treatment in this context. Revised text reads as follows:

Few data are currently available on biopsy-proven ICIM cases. In a single-center cohort study including 28 patients with clinically suspected ICIM, EMB showed signs of myocarditis/inflammation in 18 cases. ICIM patients commonly presented with dyspnea and weakness, and less frequently with symptoms of heart failure; additionally, ECG abnormalities, non-sustained ventricular arrhythmias and troponin elevation were reported. Patients’ characterization through EMB had significant therapeutic implications, since a substantial proportion of patients with low-grade myocardial inflammation were able to continue ICI treatment (new reference number [36]) (lines 177-184).

With regards to myocarditis cases temporally associated with Sars-Cov-2 infection or mRNA vaccines, we have emphasized the current lack of evidence on biopsy-proven forms and provided recent data on the safety of Sars-Cov-2 vaccination in patients with prior myocarditis. Revised text reads as follows:

To date, no definitive evidence of biopsy-proven myocarditis caused by Sars-Cov-2 infection or vaccination has been found. Conversely, a study demonstrated that Sars-Cov-2 vaccination is safe for patients with prior myocarditis; additionally, this study showed an absence of increased risk of myocarditis relapse after Sars-Cov-2 infection in the same patients’ population (new reference number [21]) (lines 148-152).

Round 2

Reviewer 3 Report

Comments and Suggestions for Authors

Revised manuscript was finely corrected.

Comments on the Quality of English Language

No comment